# Structured Illumination Microscopy of Mitochondrial in Mouse Hepatocytes with an Improved Image Reconstruction Algorithm

**DOI:** 10.3390/mi14030642

**Published:** 2023-03-12

**Authors:** Kai Hu, Xuejuan Hu, Ting He, Jingxin Liu, Shiqian Liu, Jiaming Zhang, Yadan Tan, Xiaokun Yang, Hengliang Wang, Yifei Liang, Jianze Ye

**Affiliations:** 1Sino-German College of Intelligent Manufacturing, Shenzhen Technology University, Shenzhen 518118, China; 2Laboratory of Advanced Optical Precision Manufacturing Technology of Guangdong Provincial Higher Education Institute, Shenzhen Technology University, Shenzhen 518118, China; 3College of Physics and Photoelectric Engineering, Shenzhen University, Shenzhen 518060, China; 4College of Pharmacy, Shenzhen Technology University, Shenzhen 518118, China; 5College of Physics Science and Technology, Guangxi Normal University, Guilin 541001, China

**Keywords:** super-resolution, SIM, artifact, notch filter, hepatocyte, mitochondria, lipid droplets

## Abstract

In this paper, a structured illumination microscopy (SIM) image reconstruction algorithm combined with notch function (N-SIM) is proposed. This method suppresses the defocus signal in the imaging process by processing the low-frequency signal of the image. The existing super-resolution image reconstruction algorithm produces streak artifacts caused by defocus signal. The experimental results show that the algorithm proposed in our study can well suppress the streak artifacts caused by defocused signals during the imaging process without losing the effective information of the image. The image reconstruction algorithm is used to analyze the mouse hepatocytes, and the image processing tool developed by MATLAB is applied to identify, detect and count the reconstructed images of mitochondria and lipid droplets, respectively. It is found that the mitochondrial activity in oxidative stress induced growth inhibitor 1 (OSGIN1) overexpressed mouse hepatocytes is higher than that in normal cells, and the interaction with lipid droplets is more obvious. This paper provides a reliable subcellular observation platform, which is very meaningful for biomedical work.

## 1. Introduction

The resolution of ordinary optical microscope cannot exceed the optical diffraction limit [1], which limits the observation ability of the microscope in the microscopic field. Super-resolution technology breaks through the limitation of the optical diffraction limit, and the imaging resolution is less than 200 nm, or even smaller by specific methods. The optical imaging system combined with super-resolution technology has become an indispensable observation tool in the biomedical field. At present, there are three methods to achieve super-resolution microscopy: Single Molecule Location Microscopy (SMLM) [2,3], Stimulated Emission Depletion Microscopy (STED) [4] and Structured Illumination Microscopy (SIM) [5]. The main principle of SMLM is to use a single emitter to randomly activate or switch in a continuous acquisition through a small subset of a specific dye or activated fluorescence. This method observes single-molecule switching events that are sparse enough to be identified, and then continuously collects the activated fluorescent molecules for super-resolution. The main principle of STED is to use the inhibitory effect of the loss light on the excitation light to greatly reduce the effective excitation area of the focal plane [6,7], then scan the sample, and obtain the super-resolution image through image reconstruction. Based on the principle of Moiré effect, SIM modulates the sample through the structural fringes with known frequency and phase, collects and processes the mixing signal, demodulates the frequency signal of the sample, and then expands the spectrum of the sample in the frequency domain. Compared with other super-resolution techniques, SIM has the characteristics of fast imaging speed, low phototoxicity and simple sample preparation. It has become a powerful tool for living cell biomedical research. Therefore, many research results have been proposed [8,9,10,11,12,13,14,15,16,17].

Due to the difference between theory and actual experiment, the modulation of structured light cannot ensure that its frequency and phase are exactly the same as the theoretical value. Therefore, the estimation of the modulation frequency and phase of structured light fringes have become a key step in SIM reconstruction. So far, many researchers have proposed frequency phase estimation algorithms to make the estimation of reconstruction parameters more accurate, thereby improving the quality of image reconstruction. In 2009, Shroff proposed the phase of peaks algorithm (POP) [18]. This algorithm could correctly calculate the phase when the modulation contrast of the structural fringe was high enough and the spatial frequency was moderate, but its universality was poor due to the limitation of conditions. In 2013, Wicker proposed a cross-correlation iterative minimization algorithm that could provide correct phase estimation in most cases [19]; however, this method was complex in calculation and had modulation restrictions. In the same year, Wicker proposed the autocorrelation iterative method. This method simplified the calculation, but the effect was poor when the image signal-to-noise ratio was poor [20]. In 2017, Zhang proposed an algorithm for normalizing coefficients by calculating the modulation factor and initial phase of the illumination mode and obtained a better reconstruction resolution [21]. In 2018, Cao proposed a phase estimation algorithm based on inverse matrix [22], which could obtain the analytical solution of the phase without iteration. This method could also obtain the correct phase in the case of low modulation depth. In 2023, Qian proposed an efficient robust SIM algorithm based on principal component analysis, which achieved more accurate parameter estimation and anti-noise ability than traditional cross-correlation algorithm [23].

When the high and low frequency components are separated, it is necessary to superimpose the separated frequency spectrum. Through a reasonable reconstruction algorithm to deal with the frequency spectrum of the overlapping part, the image artifacts can be reduced. In 2008, the generalized Wiener reconstruction algorithm was proposed, eight years after Gustafsson first proposed the concept of structured light super-resolution imaging [24]. This algorithm realized the lateral super-resolution for the first time and became a classical algorithm in this field. In 1972, Richardson proposed an image reconstruction algorithm based on Bayesian iteration [25]. In 1974, Lucy proposed an iterative technique of observation step-by-step correction [26]. Based on these two methods, researchers proposed the R-L reconstruction method named by two researchers. The algorithm could make the image clearer after a certain number of iterations. In 2014, Chu proposed a TV reconstruction method [27]. The reconstruction algorithm required less signal and thus reduced the photobleaching of the sample, and successfully improved the temporal resolution of the two-dimensional SIM by 15 times while maintaining the spatial resolution and image quality comparable to the traditional method. In 2017, Dan proposed a super-resolution reconstruction algorithm for optical slice images [28]. After the analysis of the imaging principle, the defocused light in the imaging process was suppressed, and the signal-to-noise ratio of the image was improved, but the imaging resolution was suppressed to a certain extent. In 2018, Huang proposed a structured illumination deconvolution algorithm based on Hessian matrix [29], which used the temporal continuity of biological samples as a priori knowledge to guide image reconstruction, and performs rolling reconstruction of the original image. At the imaging frequency of 188 Hz, the spatial resolution was increased to 88 nm, which greatly improved the spatial and temporal resolution of SIM. In 2019, Fei proposed a partial spectrum method [30], which only needed 4 original images for reconstruction but did not improve the spatial resolution. In 2021, Zhao proposed a sparse deconvolution algorithm [31], which used the prior knowledge of the sparsity and continuity of biological structures to increase the resolution of super-resolution microscopy by nearly two times. In 2022, Wang proposed a rapid reconstruction algorithm termed joint space and frequency reconstruction. By replacing Fourier domain operation with real space operation, the reconstruction speed is 80 times faster than Wiener reconstruction without losing image quality [32]. The methods mentioned above have improved the imaging quality in different aspects, but they are not good at suppressing defocused light in the frequency domain.

The problem of artifacts in SIM super-resolution image reconstruction has always been a difficulty in this research field. In the supplementary document of the article, Huang made a comprehensive summary of the types and causes of artifacts in the current SIM super-resolution image reconstruction. When the sample moves too fast, motion artifacts will be generated. When the frequency and phase estimation of the modulated wave vector is inaccurate, ring artifacts will be generated. When the Optical Transfer Function OTF measurement of the optical system is inaccurate, snowflake artifacts will be generated. When there is a defocused background in the sample image, streak artifacts will be generated. The above artifacts can be overcome in different ways. Improving the image acquisition speed of the system can overcome the motion artifacts, accurately estimating the modulation parameters can overcome the ring artifacts and accurately measuring the OTF of the system can avoid snowflake artifacts. Streak artifacts can be removed by suppressing the defocusing background. The imaging depth by the total internal reflection imaging method is very shallow, so there is no defocusing background during data acquisition. At the same time, its imaging depth limits its observation of thick samples. Therefore, the method of using algorithms to suppress is more universal than the method of adjusting optical devices aiming to solve the problem of streak artifacts caused by defocusing background during thick sample imaging. However, many existing algorithms are not effective in suppressing such artifacts. As shown in Figure 1 below, the streak artifacts in the circle area in the figure obviously lead to low signal-to-noise ratio of the image. Therefore, this paper proposes a structured light super-resolution image reconstruction algorithm combining Notch function (Notch SIM, N-SIM for short) for the streak artifacts caused by the defocused background.

By analyzing the mathematical model of structured light illumination imaging, a filter was designed to process the low-frequency signal in the imaging process. Under the condition of retaining the information of the image, the streak artifacts caused by defocused light in the imaging process were effectively suppressed, and the imaging quality of the image was improved. Mitochondria are organelles responsible for producing energy in organisms. They are the main place for cells to carry out aerobic respiration and are closely related to the state of cells. Long-term damage to mitochondria is shown to be associated with the occurrence of liver disease, and changes in mitochondrial function is confirmed in a variety of chronic liver diseases. Studies have shown that OSGIN1 allele imbalance is associated with the progression of liver cancer [33]. The N-SIM algorithm proposed in this paper was applied to reconstruct the mitochondria and lipid droplets in mouse hepatocytes, and then the MATLAB development tool was developed to detect and identify the images. It was found that the mitochondria in OSGIN1 overexpressed cells were more active, and the interaction with lipid droplets was more obvious. This finding is very meaningful, indicating that the SIM system combined with N-SIM algorithm mentioned in this paper can provide a clear subcellular organelle observation platform for biomedical workers, which can visualize metabolic diseases.

## 2. Materials and Methods

The optical imaging system of a wide-field microscope was actually equivalent to a low-pass filter. Due to the limitation of optical diffraction, the process of imaging the sample actually truncated the high-frequency information (the fine structure of the sample) of the sample beyond the support domain of the optical transfer function (OTF), thereby limiting the spatial resolution of the imaging. The SIM could move high-frequency information to the OTF support domain to achieve super-resolution. The imaging principle is shown in Figure 2.

The illumination light with spatial structure excited the sample in the SIM. Based on the principle of Moiré effect, the high frequency information of the sample (the detail information of the sample) that cannot be obtained under the wide-field illumination mode was modulated into the OTF support domain of the optical microscopy imaging system. Then, through the frequency domain signal demodulation, high and low frequency signal separation, shift, reconstruction and other steps could achieve a relative wide-field microscope imaging twice the resolution. In SIM, the cosine structured light is usually applied to illuminate the sample. The Fourier transform of the spectral expression of the SIM raw image is as follows:(1)$${\tilde{E}}_{d,\varphi }\left(k\right)=I_{0}\left[\tilde{C}\left(k\right)+\frac{m}{2}\tilde{C}\left(k+p_{d}\right)\exp\left(-i\varphi \right)+\frac{m}{2}\tilde{C}\left(k-p_{d}\right)\exp\left(i\varphi \right)\right]\tilde{H}\left(k\right)+{\tilde{D}}_{bg}\left(k\right)$$
where $\tilde{H}(k)$ is the OTF of the optical system, $\tilde{D_{bg}}(k)$ represents the defocus signal, i is the imaginary unit, d represents the direction of the structured illumination light, φ represents the initial phase of the structured illumination light. The spectrum of the original image contained not only the low frequency component $\tilde{C}(k)\tilde{H}(k)$, but also the high frequency components $\tilde{C}(k+p_{d})\tilde{H}(k)$ and $\tilde{C}(k-p_{d})\tilde{H}(k)$. $\tilde{C}(k)$ is the low-frequency information of the sample after the deconvolution operation. $\tilde{C}(k+p_{d})$ and $\tilde{C}(k-p_{d})$ are high-frequency information of samples after deconvolution. The spectrum of this part contained the detailed (super-resolution) structure information of the sample. Since the OTF of the system was actually constant, the structured light illumination method using cosine mode was also limited by diffraction. Therefore, in general, the modulation frequency $p_{d}$ of the cosine structured light would not exceed the cut-off frequency $k_{em}$ of the system OTF, so the highest frequency of the sample that the system could observe in this case was $p_{d}+k_{em}$.

Through the above methods, the resolution in one direction can be improved. If the super-resolution of the image in the whole plane is needed, the structural fringes need to be rotated 60° and 120°, respectively, for the same operation. By obtaining three original images with a phase interval of 2π/3, the equations could be constructed. As shown in Figure 2, the high-frequency components and low-frequency components in each direction could be obtained by solving the following equations. As shown in Formula (2):(2)$$\left[\begin{array}{c} {\tilde{E}}_{d,\varphi 1}\left(k\right)\\ {\tilde{E}}_{d,\varphi 2}\left(k\right)\\ {\tilde{E}}_{d,\varphi 3}\left(k\right) \end{array}\right]=I_{0}\left[\begin{array}{ccc} 1 & \frac{m}{2}\exp\left(-i\varphi_{1}\right) & \frac{m}{2}\exp\left(i\varphi_{1}\right)\\ 1 & \frac{m}{2}\exp\left(-i\varphi_{2}\right) & \frac{m}{2}\exp\left(i\varphi_{2}\right)\\ 1 & \frac{m}{2}\exp\left(-i\varphi_{3}\right) & \frac{m}{2}\exp\left(i\varphi_{3}\right) \end{array}\right]\cdot \left[\begin{array}{c} \tilde{C}\left(k\right)\tilde{H}(k)\\ \tilde{C}\left(k+p_{d}\right)\tilde{H}(k)\\ \tilde{C}\left(k-p_{d}\right)\tilde{H}(k) \end{array}\right]+\left[\begin{array}{c} {\tilde{D}}_{{bg}_{d,\varphi 1}}\left(k\right)\\ {\tilde{D}}_{{bg}_{d,\varphi 2}}\left(k\right)\\ {\tilde{D}}_{{bg}_{d,\varphi 3}}\left(k\right) \end{array}\right]$$

In general, the observation sample of the optical microscope was thin after special preparation. However, the samples used in dynamic imaging of living cells were thick, this inevitably produced defocus noise. If the defocus signal was not suppressed, it would have produced streak artifacts in the process of image reconstruction [28], affecting the imaging quality. The schematic diagram of the imaging process is shown in Figure 3.

From the Equation (1), it can be seen that the low frequency signal and defocus signal of the collected sample had the same frequency range, so the Equation (1) can be rewritten as follows:(3)$${\tilde{E}}_{d,\varphi }\left(k\right)=I_{0}\left[\frac{I_{0}\tilde{H}\left(k\right)\tilde{C}\left(k\right)+{\tilde{D}}_{bg}\left(k\right)}{I_{0}\tilde{H}\left(k\right)}+\frac{m}{2}\tilde{C}\left(k+p_{d}\right)\exp\left(-i\varphi \right)+\frac{m}{2}\tilde{C}\left(k-p_{d}\right)\exp\left(i\varphi \right)\right]\tilde{H}\left(k\right)$$

Therefore, the defocus signal only affected the low frequency spectrum of the sample spectrum and did not affect the high frequency signal. In the frequency domain, the defocus signal and the low frequency spectrum were mixed together, so the appropriate processing of the low frequency spectrum could achieve better defocus signal suppression.

In this paper, based on the generalized Wiener reconstruction algorithm, the reconstruction algorithm expression was rewritten appropriately. A notch function was added to the equation to limit the defocus signal in the low frequency spectrum. The generalized Wiener reconstruction expression is as follows:(4)$$g\left(r\right)=ifft\left(\frac{\sum_{n,d}{\tilde{H}}_{nd}^{*}\left(k+np_{d}\right)S_{nd}\left(k+np_{d}\right)}{\sum_{n}{\left|{\tilde{H}}_{nd}\left(k+np_{d}\right)\right|}^{2}+\alpha^{2}}\right)A(\tilde{k})$$
where g(r) is the final super-resolution reconstruction image, ${\tilde{H}}^{*}(k)$ is the conjugate of the optical transfer function in the direction of each structural illumination stripe, $S_{nd}\left(k+np_{d}\right)$ is the high and low frequency spectrum component, when n is equal to 0, it is the low frequency spectrum, when n is equal to ±1, it is the high frequency spectrum, α is the Wiener parameter, ifft is the inverse Fourier transform operation, $A(\tilde{k})$ is the apodization function, the Equation (4) is rewritten as follows:(5)$$g\left(r\right)=ifft\left(\frac{{\tilde{H}}_{0d}^{*}(k)S_{0d}(k)}{{\left|{\tilde{H}}_{0d}\left(k\right)\right|}^{2}+\alpha^{2}}F(k)+\frac{\sum_{n=\pm 1,d}{\tilde{H}}_{nd}^{*}\left(k+np_{d}\right)S_{n}\left(k+np_{d}\right)}{\sum_{n=\pm 1,d}{\left|{\tilde{H}}_{nd}\left(k+np_{d}\right)\right|}^{2}+\alpha^{2}}\right)A(\tilde{k})$$
where F(k) is a notch function introduced to suppress the low frequency spectrum. By controlling the parameters of F(k), the effective spectrum information can be retained and the defocus signal in the low frequency spectrum can be suppressed. The function form of the notch filter mentioned in this paper is as follows:(6)$$g(x,y)=\frac{1}{2\pi \sigma^{2}}e^{-\frac{\left(x^{2}+y^{2}\right)}{2\sigma^{2}}}$$

The above Equation (6) is the expression of the two-dimensional Gaussian function, which can be transformed to a notch function by the following Equation (7), and the notch degree can be controlled by controlling the parameter σ of the Gaussian function.
(7)$$F\left(k\right)=\frac{(\max\left\{g\left(x,y\right)\right\}-g(x,y))}{\max\{g(x,y)\}}$$
where max represents the maximum value of Gaussian function.

## 3. Result

### 3.1. Simulation Analysis of N-SIM Algorithm

In order to verify the N-SIM algorithm, this paper used the resolution board to analyze the algorithm. First, we added Gaussian noise with mean value of 0.1 and variance of 0.1 to the original image, then simulated the modulation of the image, and then used Wiener reconstruction algorithm, RL reconstruction algorithm, HIFI reconstruction algorithm and N-SIM algorithm to reconstruct the modulated image, respectively. The result of image reconstruction is shown in Figure 4 below. The red square area is a magnified view of the corresponding position. Because the structure of the original image was damaged in the process of adding Gaussian noise, the reconstructed image was slightly deformed, and we actually used the image with Gaussian noise as the original image for analog modulation. The intensity of the image reconstructed with N-SIM in the dark area was closer to 0, and the overall gray intensity was closer to the original image. By analyzing the peak signal to noise ratio (PSNR) and structure similarity (SSIM), N-SIM had better performance.

### 3.2. Analysis of Suppressing Effect of N-SIM Algorithm on Defocused Light

A two-dimensional structured light super-resolution imaging system was manually built in this study, as shown in Figure 5. The incident laser was diffracted by a spatial light modulator, and the ±1-order diffracted light was selected to interfere on the surface of the sample to form a structural excite in the sample, with a 100-fold objective lens (OLYMPUS 100×/1.45 Oil) and a 60-fold objective lens (OLYMPUS 60×/1.5 Oil). A Kuro CMOS camera (pixel size = 11 μm) was employed for data acquisition. A mitochondrial green, fluorescent probe (Mito-Tracker Green, excitation wavelength 490 nm) was used to stain mitochondria in mouse hepatocytes. Mitochondria were excited by a 488 nm laser and data were collected from a 60-fold objective lens (NA = 1.5).

After the original image was collected, the generalized Wiener-SIM, RL-SIM, HIFI-SIM and N-SIM algorithm were applied to reconstruct the original image. The reconstruction result is shown in Figure 6. It can be seen from the results that the generalized Wiener-SIM, RL-SIM and HIFI-SIM produced stripe-like reconstruction artifacts due to the inability to suppress the defocusing light in the original image, which seriously affected the reconstruction quality of the image. It can be found from Figure 6 that the improved N-SIM algorithm achieved better suppression of the defocus signal of the image, so the background was relatively pure.

The fluorescence intensity of the four reconstructed images was analyzed. The results are shown in Figure 7 below. It can be seen that the background of the image reconstructed by N-SIM algorithm was lower than that of the other three algorithms, and the quality of the reconstructed image was higher than that of the other three reconstruction methods.

In order to verify that N-SIM algorithm has more obvious suppression effect on background noise, the fluorescence intensity fluctuation on the yellow line (the yellow line only contains the fluorescence information of the background area) of the background area in Figure 6a was analyzed. The results are shown in Figure 8 below. It can be found that the fluorescence intensity fluctuation of the image reconstructed with N-SIM algorithm at its background position was smaller, and the overall range was close to zero. Therefore, N-SIM algorithm had a better suppression effect on the defocused background than the other three reconstruction algorithms.

### 3.3. Validation of N-SIM

Since the principle of this method was realized by suppressing the low-frequency spectrum of the sample, it might have affected the effective information (sample structure information) of the sample. In order to verify the effectiveness of this method, an adjustable parameter was given in this paper to control the suppression of the notch function on the low-frequency of the center position. When setting different values, different results would be obtained. It can be seen from Figure 9 that the red arrow points to a mitochondrion with weak signal strength. When σ = 2, the whole image could not only suppress the background noise, but also maintain the effective information of the samples.

### 3.4. System Resolution Calibration

The system calibration was carried out by imaging 50 nm fluorescent microbeads. The image reconstruction was performed with the N-SIM algorithm as shown in Figure 10.

The full width at half maximum (FWHM) of the reconstruction algorithm corresponding to the imaging result was calculated by the gray value statistics of the pixels on the yellow line position in Figure 10, and the result is shown in Figure 10c,d. After calculation, the full width at half maximum of 50 nm fluorescent microspheres was 295.13 nm under the condition of wide-field imaging. After image reconstruction with N-SIM algorithm, the FWHM of fluorescent microspheres was 143.6 nm. Therefore, the imaging resolution of the system in this paper reached 144 nm.

## 4. Discussion

OSGIN1 gene has a strong tumor suppressor function, mainly encoding oxidative stress proteins involved in the regulation of cell death [34]. Therefore, it is of great significance to study the expression of OSGIN1 gene in cells to overcome the disease. The sample cells in this paper were OSGIN1 gene overexpression cells, and normal cells are control groups for experiments. The difference between mitochondria and lipid droplets in OSGIN1 gene overexpression cells and normal mouse hepatocytes is explored in this chapter.

### 4.1. Sample Preparation and Data Acquisition

The sample cells in this paper were purchased from the American culture library (Rockville, MD, USA), cultured in a basal medium supplemented with 10% fetal bovine serum and ITS-G (5 mg/mL insulin, 5 mg/L transferrin, 5 μg/L selenite), cultured in an incubator containing 5% carbon dioxide (37 °C), and using oleic acid and palmitic acid in 75% ethanol. The mixture was heated to 55 °C and completely dissolved for membrane formation. Lipid droplets and mitochondria in mouse hepatocytes were then stained with Bodipy493/503 and Mito-tracker Red CMXRos, respectively. Membrane formation and staining were performed in a confocal dish.

Sample data acquisition through the optical system mentioned above. The CMOS camera collected 9 original images, the sampling interval was 5 ms, and a set of pictures is collected every 30 s. Then the N-SIM algorithm mentioned in the paper was used to reconstruct the image to achieve image super-resolution. After reconstruction, a dynamic image with a time resolution of 30 s could be obtained.

### 4.2. Data Analysis

The OSGIN1 gene overexpressed cells and normal cells were subjected to structural illumination imaging, and the collected data were super-resolution reconstructed with the N-SIM algorithm. The time interval between each frame of images was 30 s. The mitochondria in each frame of images were observed. The results are shown in Figure 11.

It can be observed from Figure 11 that the mitochondria in normal mouse hepatocytes had less morphological changes from 0 s to 210 s, while the cells with OSGIN1 gene overexpression had more frequent mitochondrial morphological changes over time, resulting in mitochondrial fission and fusion. Applying the tool (mainly using image enhancement, connected domain analysis, circle detection and other methods, which are implemented by us through MATLAB) identified and counted the mitochondria in the collected cell images, the results are shown in Figure 12a,b. From the diagram, it can be seen that the number of mitochondria in normal cells changed more smoothly than the number of mitochondria in OSGIN1 gene overexpressing cells, indicating that the number of mitochondrial fusion and fission events in OSGIN1 gene overexpressing cells was more, so the mitochondrial activity of OSGIN1 gene overexpressing cells was stronger. From the results of immunoblotting in Figure 12c, the specific proteins in the same number of normal cells and OSGIN1 gene overexpressed mouse hepatocytes were analyzed, respectively. GAPDH was an intracellular reference, which could ensure the same number of cells in the experiment. Under the condition of the same number of cells, the OSGIN1 protein contained in the OSGIN1 gene overexpressed cells was more than the same protein in the normal cells, which was in line with the logic of the experiment. In this case, we tested the PPIF (mitochondrial-specific protein) in the sample and found that the total amount of mitochondrial proteins in the two cells was basically the same. Then, the lipid droplet protein (PLIN2) on mitochondria was detected. It was found that the lipid droplet protein on mitochondria of OSGIN1 gene overexpression cells was significantly more than that of ordinary cells, which indicated that the interaction between mitochondria and lipid droplets in OSGIN1 gene overexpression cells was more active.

In hepatocytes, mitochondria interact with lipid droplets, so the activity of mitochondria also has a great influence on the state of lipid droplets in cells. The samples were observed using a 100× objective lens. After image reconstruction, it was found that there were more and larger lipid droplets in normal mouse hepatocytes, and the morphology of lipid droplets in OSGIN1 gene overexpressed cells was smaller and less, as shown in Figure 13a,b.

In this paper, MATLAB software was applied to realize lipid droplet recognition and size detection. The size distribution of lipid droplets in normal mouse hepatocytes and OSGIN1 gene overexpressed hepatocytes was obtained as shown in Figure 13c. It could be found that there are lipid droplets with a radius of more than 1.1 μm in normal cells, while the radius of lipid droplets in OSGIN1 overexpressed cells was all less than 0.9 μm, and the number of lipid droplets with a radius of 0.9 μm in normal cells was significantly higher than that in OSGIN1 overexpressed cells.

### 4.3. Phenomenon Analysis

Via data collection and reconstruction, it was found that OSGIN1 gene overexpressed cells had stronger mitochondrial activity. Specifically, during the same time, the number of mitochondria in OSGIN1 gene overexpressed mouse hepatocytes changed greatly, while the number of mitochondria in normal mouse hepatocytes was relatively more stable. It was also shown that there were more mitochondrial fission and fusion events in OSGIN1 gene overexpressing cells than in the normal cells.

In addition, mitochondria in OSGIN1 overexpressed hepatocytes significantly promoted the decomposition of lipid droplets. This is evidenced by more and larger lipid droplets in normal hepatocytes than in OSGIN1 overexpressed hepatocytes. This observation is further confirmed by biological experiments, which verify the reliability of the system and N-SIM algorithm mentioned in this paper.

## 5. Conclusions

In this paper, a Wiener reconstruction algorithm combined with notch function is proposed by analyzing the principle of generalized Wiener reconstruction algorithm. Through the processing of the low frequency part in the reconstruction process, the streak artifacts caused by defocused light in the image reconstruction process are successfully controlled, and the lateral resolution of 144 nm is realized, which improves the imaging quality of image reconstruction. The structured light super-resolution image reconstruction software is developed by MATLAB, which integrates the two existing reconstruction algorithms and the reconstruction algorithm proposed in this paper. In the experiment, it is found that the activity of mitochondria in OSGIN1 gene overexpressed cells is higher than that in normal cells, and the decomposition of lipid droplets is more obvious. For the two organelles of mitochondria and lipid droplets, an image recognition detection program is developed by MATLAB.

The N-SIM algorithm proposed in this paper suppresses the defocused light in the imaging process. However, the imaging resolution of the system is not improved, and the reconstruction time of the algorithm is long (512 × 512 pixels, CPU i5-10300H, GTX 1650 4 GB, RAM 16 GB, the whole calculation process takes about 9 s). Camera with smaller pixels can be applied, and the algorithm process can be optimized to improve the efficiency of software operation.

In summary, this paper provides a stable and reliable observation platform for biomedical researchers for mitochondria and lipid droplets in biological cells.

## Figures and Tables

**Figure 1 micromachines-14-00642-f001:**
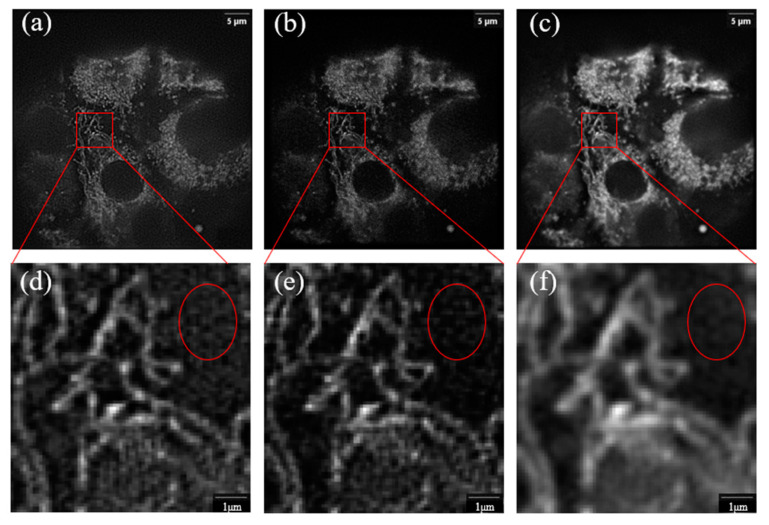
(**a**–**c**), respectively, use wiener reconstruction [33], RL reconstruction [25,26], HIFI reconstruction [31] to reconstruct the mitochondria of mouse liver cells collected under the 60× objective lens; (**d**–**f**) are magnified images of square areas in (**a**–**c**).

**Figure 2 micromachines-14-00642-f002:**
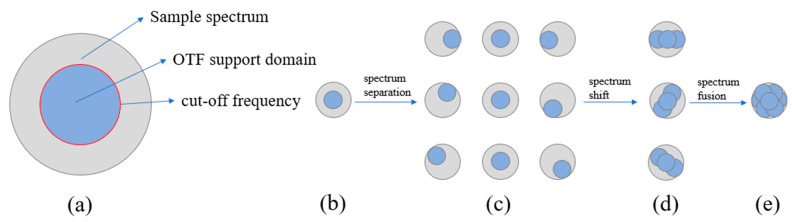
(**a**) frequency detection range of fluorescence microscopy systems of wide-field illumination; (**b**) image spectrum acquired in wide-field illumination mode; (**c**) the 0°, 60° and 120° directions are used respectively, and the initial phase difference in each direction is 2π/3 of the sinusoidal mode structure stripe light illumination sample. The high and low frequency spectrum components are obtained by spectrum separation of the collected images; (**d**) spectral fusion in a single direction can achieve image super-resolution in a single direction; (**e**) by fusing the high and low frequency spectrum in each direction, the super-resolution image of the two-dimensional plane can be obtained.

**Figure 3 micromachines-14-00642-f003:**
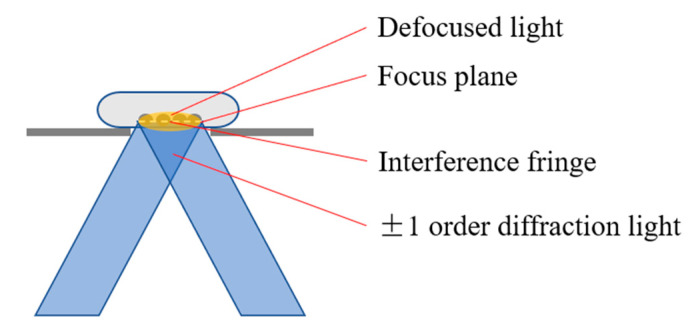
In the imaging process of the fluorescence inverted microscope, the ±1 order diffraction light interferes on the focal plane to generate stripe structured light and then illuminate the sample. At this time, in addition to the focal plane being excited, a small part of the defocus signal is also excited.

**Figure 4 micromachines-14-00642-f004:**
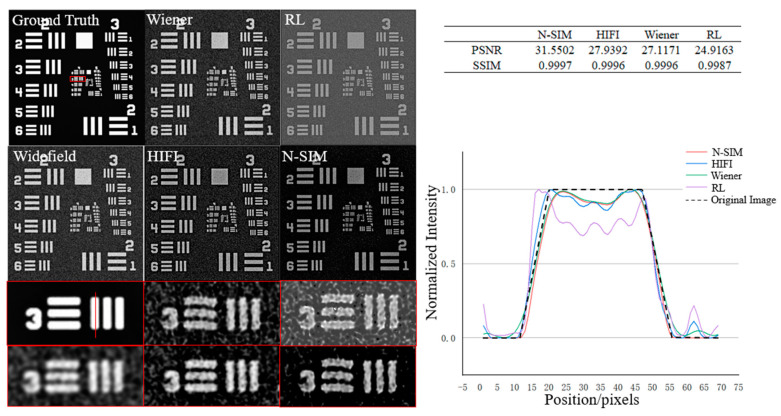
Reconstruction results of modulated resolution board using Wiener-SIM, RL-SIM, HIFI-SIM, N-SIM respectively, and gray value analysis results on the red line. The images below are magnified images corresponding to the red box area above.

**Figure 5 micromachines-14-00642-f005:**
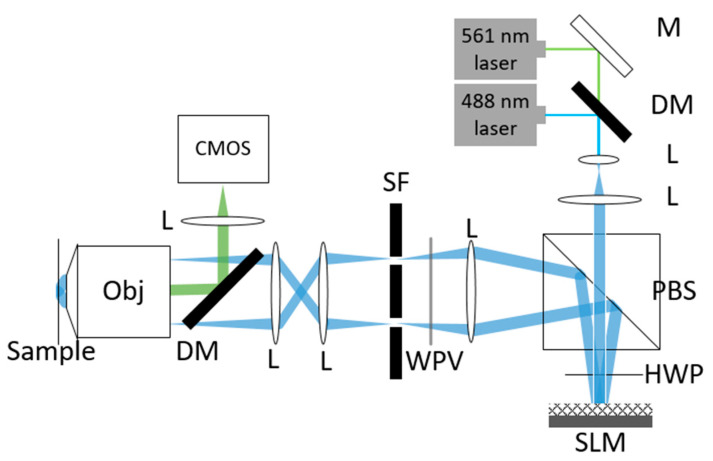
Structured light super-resolution imaging system built in this paper. M is a mirror, DM is a dichroic mirror, L is an achromatic lens, PBS is a polarization beam splitter, HWP is an achromatic 1/2 wave plate, SLM is a spatial light modulator, WPV is a zero-order vortex half-wave plate, SF is a spatial light filter, MO is a microscopic objective lens, CMOS is a detection camera.

**Figure 6 micromachines-14-00642-f006:**
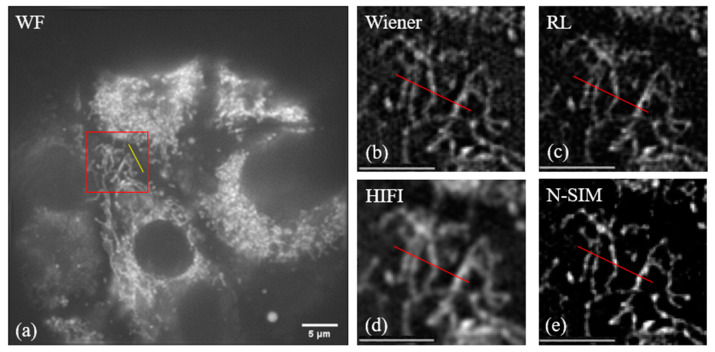
(**a**) a 60-fold objective lens is employed to collect the wide-field map of the sample; (**b**) imaging result reconstructed by Wiener-SIM; (**c**) imaging result reconstructed by RL-SIM; (**d**) imaging result reconstructed by HIFI-SIM; (**e**) imaging result reconstructed by N-SIM; (**b**–**e**) are magnified images of the square area in (**a**) with these four methods. The scale length in (**b**–**e**) is 5 μm.

**Figure 7 micromachines-14-00642-f007:**
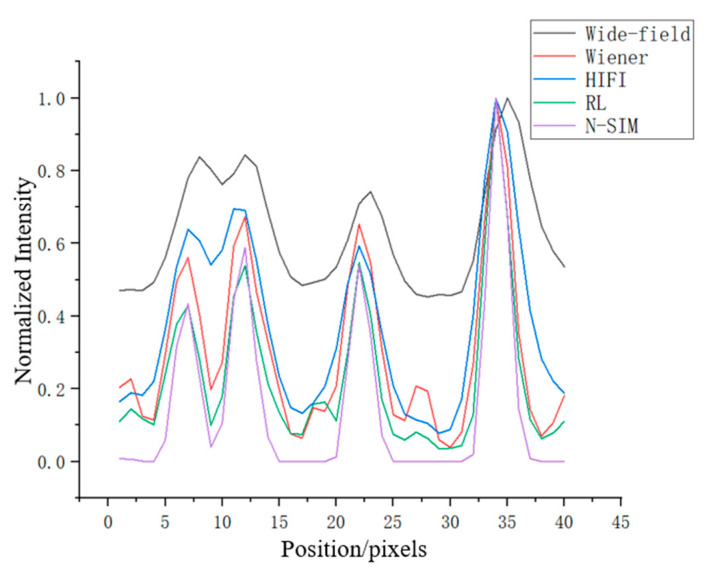
Fluorescence intensity statistics at the red line position in Figure 5.

**Figure 8 micromachines-14-00642-f008:**
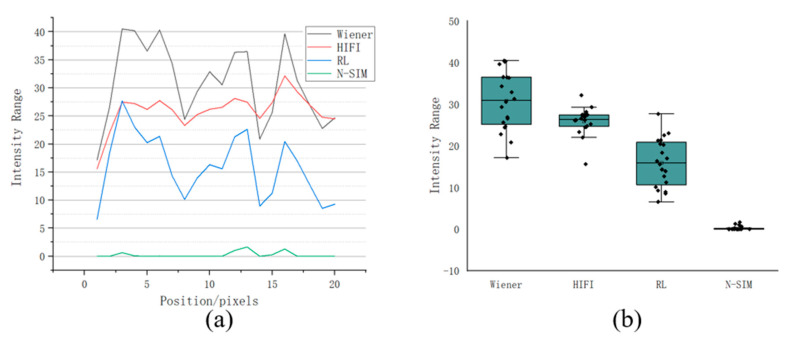
(**a**) Wiener-SIM, HIFI-SIM, RL-SIM and N-SIM are respectively used to reconstruct the sample image, and the fluctuation of fluorescence intensity on the yellow line in the background area is analyzed. (**b**) The distribution of fluorescence intensity of each pixel in the yellow line.

**Figure 9 micromachines-14-00642-f009:**
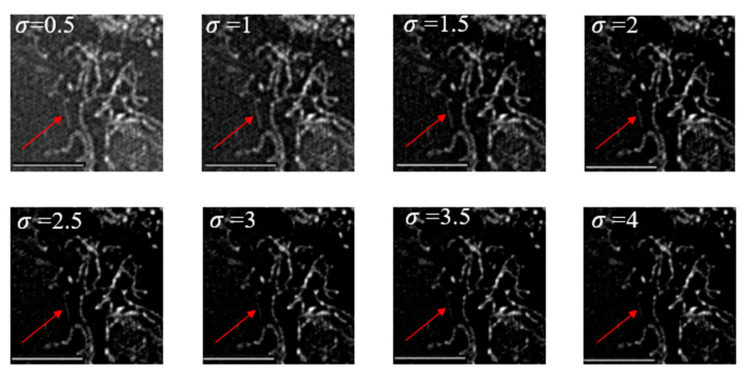
Under the 60× objective lens, when σ takes different values, N-SIM is used to reconstruct the image of mouse hepatocyte mitochondria. The length of the image scale is 5 μm.

**Figure 10 micromachines-14-00642-f010:**
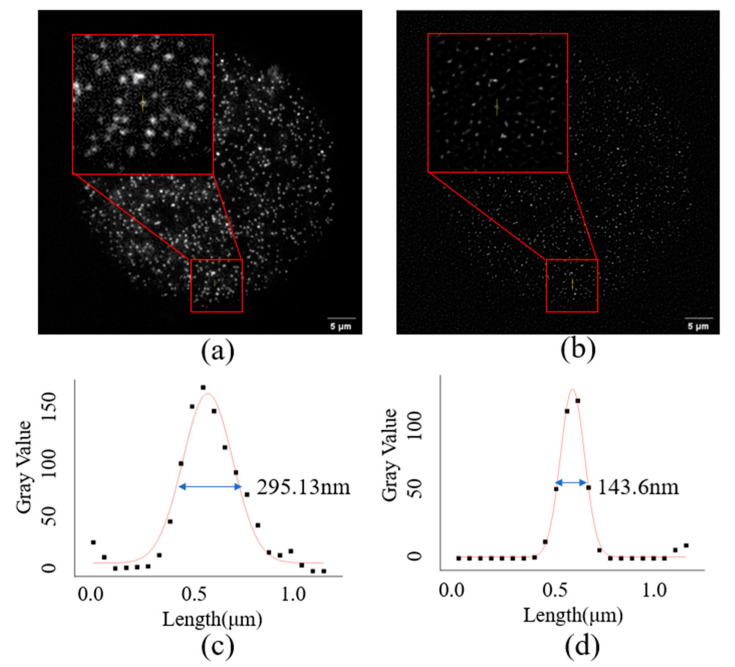
Imaging results of 50 nm fluorescent beads. (**a**) the result of wide-field imaging; (**b**) the imaging result after image reconstruction with N-SIM. The red line part in the figure is the enlarged figure of the corresponding position; (**c**) the full width at half maximum (FWHM) of 50 nm fluorescent microspheres under the condition of wide-field imaging; (**d**) the FWHM of 50 nm fluorescent microspheres after N-SIM image reconstruction algorithm.

**Figure 11 micromachines-14-00642-f011:**
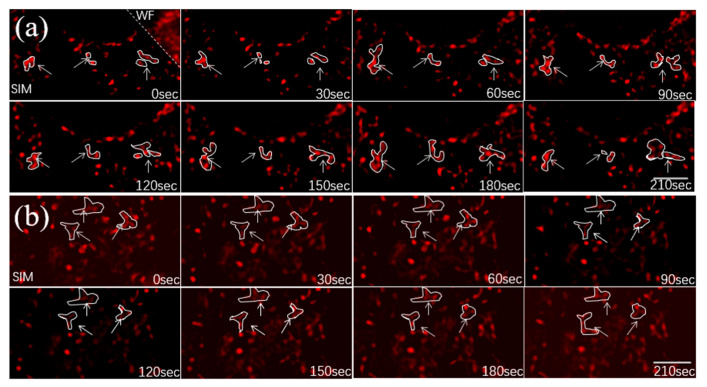
A set of original images were collected every 30 s, and the results of normal cells and OSGIN1 gene overexpression cells were reconstructed by N-SIM algorithm. (**a**) OSGIN1 gene overexpression cell; (**b**) normal cell, scale bar is 3μm.

**Figure 12 micromachines-14-00642-f012:**
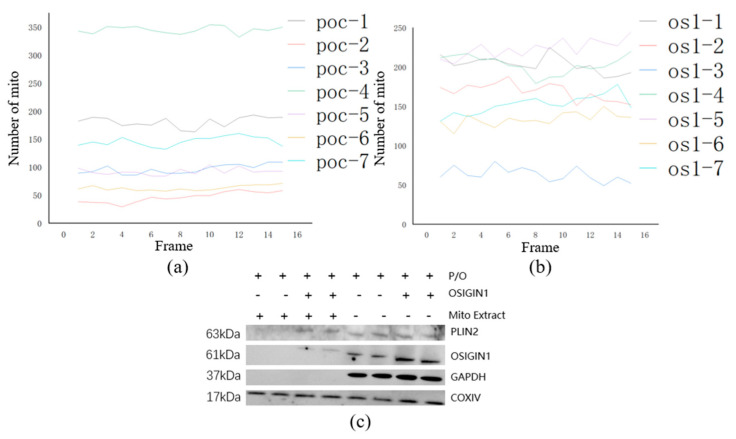
Normal mouse hepatocytes and OSGIN1 gene overexpressed cells were collected every 30 s, and mitochondrial recognition and counting were performed using MATLAB development tools. (**a**) the change trend map of the number of mitochondria in normal cells. (**b**) the change map of the number of mitochondria in OSGIN1 gene overexpression cells. (**c**) the result of immunoblotting. The samples from left to right are normal cells to extract mitochondria, OSGIN1 gene overexpression cells to extract mitochondria, normal cells, and OSGIN1 gene overexpression cells. All cells were treated with P/O under the same conditions.

**Figure 13 micromachines-14-00642-f013:**
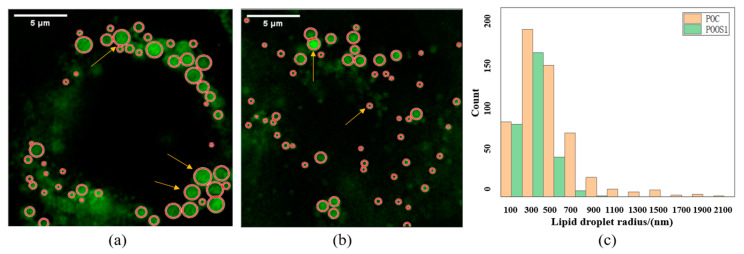
The lipid droplet images of mouse hepatocytes were collected under a 100-fold objective lens and the results were reconstructed with the N-SIM algorithm. (**a**) OSGIN1 gene overexpression cells; (**b**) normal cells; (**c**) the lipid droplets of normal mouse hepatocytes and OSGIN1 gene overexpression cells were identified and counted. A total of 884 lipid droplets were counted. Yellow column represents normal mouse hepatocytes, and green column represents OSGIN1 gene overexpression mouse hepatocytes. The transverse axis is the radius of lipid droplets, the unit is nanometer, and the longitudinal axis is the number of lipid droplets corresponding to the size.

## Data Availability

Data are available from the corresponding author upon reasonable request.

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
