# Peer review of "Structured Illumination Microscopy of Mitochondrial in Mouse Hepatocytes with an Improved Image Reconstruction Algorithm"

_micromachines, 2023, doi:10.3390/mi14030642_

Round 1

Reviewer 1 Report

I have carefully read and reviewed the manuscript entitled " Structured Illumination Microscopy of Mitochondrial in Mouse Hepatocytes with an Improved Image Reconstruction Algorithm", Hu et al. The authors present a new algorithm for processing the raw data of structured illumination microscopy (SIM), which suppresses the streak artifacts caused by defocused signals during the imaging process without losing the effective information of the image. The method is successfully applied to identify, detect and count the reconstructed images of mitochondria and lipid droplets. The structure of this work is well organized and clear and the manuscript is well written. The results are solid and demonstrated the excellent performance for reliable subcellular observation. I recommend this work to be published in micromachines. Some minor issues are listed below.

1. Regarding the previous work on SIM, the authors missed some latest advances in the introduction section, e.g. [Wang et al. High-speed image reconstruction for optically sectioned, super-resolution structured illumination microscopy. Advanced Photonics, 2022, 4(2)] and [Qian et al. Structured illumination microscopy based on principal component analysis. eLight, 2023, 3(1)]

2. The author compares the N-SIM with other three algorithms as Winner-SIM, RL-SIM and HiFi-SIM. However, the references of these algorithms are not matching with themselves in line 251. Please check the correctness of these references. Meanwhile, these references should mark at the caption of figure 1 and its relevant text, where they are first mentioned.

3. What is the yellow line in Figure 6(a)? It is in a different position from the red line in figure 6(b) and lacks description.

Author Response

Dear Reviewer,

Thank you for your patience in reviewing the manuscript. We have responded to your comments. Please see the attachment.

Reviewer 2 Report

The authors introduce a notch filtering process into a structured illumination microscopy (SIM), they call the method N-SIM, to suppress the artifact due to the object at the defocus position. The authors estimated the performance of N-SIM compared with other reconstruction methods. The N-SIM was applied to observe the activity of mitochondria in live cells of mouse hepatocytes.

The results seem well. However, presentations of the manuscript can be improved. 

Questions and comments,

1.    The function C(k) may be important in Eq.(1) and the presented principle. The authors should explain about more details of C(k).

2.    I can understand the deformation from Eq. (1) to (3). However I could not catch the meaning of the deformation. Especially, what does the first term in right hand-side mean? Why is this form necessary in the principle?

3.    In Eq.(7), the authors should clear the definition of the function of “max”. If it means the maximum value of the Gaussian function, the maximum value of Gaussian function can be analytically described.

4.    The authors estimated structure similarity (SSIM) on a red line in Figure 4. I can agree that N-SIM shows the high performance just on the red line. However, the three horizontal stripes on the left side of the red line slightly deform in Wiener and N-SIM. Could the authors give comments on the shape of a reconstructed image?

5.    In Figure 7, Winer and LR show a small peak at position = 27 pixels. This peak is not reconstructed in N-SIM. Is it a noise? If so, how did the authors distinguish noise and signal?

6.    Line 283, what does “the effective information” mean?

7.    In chapter 4, line 311-322, there is a description about the objective of this chapter. The authors should present such objective at the introduction of the paper.

8.    The authors use a MATLAB software to recognize the lipid droplet and its analysis. Is it a general method? If it is an original one by the authors, the method should be clear, else some literatures should be referred.

9.    In line 424, the authors mention about the reconstruction time. How long time is necessary for the reconstruction?

Minors

Line 109, the first appearance of “OTF” should be spelled out.

Line 238, a period “.” is necessary after “Figure 5”.

Author Response

(The authors gave the same response as above.)
